# Unraveling two distinct polymorph transition mechanisms in one n-type single crystal for dynamic electronics

Daniel William Davies [1,10], Bumjoon Seo [2,7,10], Sang Kyu Park [1,8], Stephen B. Shiring [2], Hyunjoong Chung[1], Prapti Kafle[1], Dafei Yuan[3,9], Joseph W. Strzalka [4], Ralph Weber[5], Xiaozhang Zhu [3], Brett M. Savoie [2] ✉ & Ying Diao [1,6] ✉

Cooperativity is used by living systems to circumvent energetic and entropic barriers to yield highly efficient molecular processes. Cooperative structural transitions involve the concerted displacement of molecules in a crystalline material, as opposed to typical molecule-by-molecule nucleation and growth mechanisms which often break single crystallinity. Cooperative transitions have acquired much attention for low transition barriers, ultrafast kinetics, and structural reversibility. However, cooperative transitions are rare in molecular crystals and their origin is poorly understood. Crystals of 2-dimensional qui-noidal terthiophene (2DQTT-o-B), a high-performance n-type organic semi-conductor, demonstrate two distinct thermally activated phase transitions following these mechanisms. Here we show reorientation of the alkyl side chains triggers cooperative behavior, tilting the molecules like dominos. Whereas, nucleation and growth transition is coincident with increasing alkyl chain disorder and driven by forming a biradical state. We establish alkyl chain engineering as integral to rationally controlling these polymorphic behaviors for novel electronic applications.

A class of emergent phenomenon[1], known as cooperativity, grants the bypassing of large energy barriers in a wide variety of systems, ranging from spin flipping in magnetic materials to protein folding[2–5]. In solid state phase transitions, cooperative mechanisms occur by a diffusionless displacement of molecules with each molecule retaining its nearest neighbors. This results in ultrafast kinetics and lower energy barriers, relative to nucleation and growth type mechanisms, which

occur molecule by molecule. The displacive nature of cooperativity, well documented in inorganic first-order martensitic transitions, results in unique behaviors, such as reversible shape changes in the crystal leading to thermosalient motion[6–9] and shape memory[7,10]. A key distinction from nucleation and growth rises from interactions with defects within the crystal, which results in avalanche behavior due to pinning of the phase boundary[4,11–14]. While cooperative behavior has

[1]Department of Chemical and Biomolecular Engineering, University of Illinois at Urbana-Champaign, 600 South Mathews Avenue, Urbana, IL 61801, USA. [2]Davidson School of Chemical Engineering, Purdue University, 480 W Stadium Ave, West Lafayette, IN 47907, USA. [3]Beijing National Laboratory for Molecular Sciences, CAS Key Laboratory of Organic Solids, Institute of Chemistry, Chinese Academy of Sciences, Beijing 100190, P. R. China. [4]X-Ray Science Division, Argonne National Laboratory, Argonne, IL 60439, USA. [5]Bruker BioSpin Corp., 15 Fortune Drive, Billerica, MA 01821, USA. [6]Beckman Institute for Advanced Science and Technology, 405 N. Mathews Ave. M/C 251, Urbana, IL 61801, USA. [7]Present address: Department of Chemical and Biomolecular Engineering, Seoul National University of Science and Technology, 232 Gongneung-ro, Nowon-gu, Seoul 01811, Republic of Korea. [8]Present address: Institute of Advanced Composite Materials, Korea Institute of Science and Technology, Joellabuk-do 55324, South Korea. [9]Present address: College of Materials Science and Engineering, Hunan University, Changsha 410082, China. [10]These authors contributed equally: Daniel William Davies, Bumjoon Seo. ✉e-mail: bsavoie@purdue.edu; yingdiao@illinois.edu

been well documented in metal alloy based martensitic transitions, it has rarely been reported in molecular crystals. As such, the fundamental origin of cooperativity in organic crystals is not well understood and the molecular design inducing this behavior is still unknown.

Recently, cooperative transitions have been observed in key organic semiconductor molecules such as 6,13-bis-(triisopropylsilylethnyl) pentacene (TIPS-P), among other p-type semiconductors[9,15,16]. Cooperativity in TIPS-P led to fast switching of the electronic properties[17] coupled with large recoverable crystal shape change[15], resulting in super- and ferro-elastic mechanical effects[18]. In the case of TIPS-P, the rotation of the TIPS groups played a key role in the observed cooperative phenomena[16,19]. Besides dynamic rotation, other systems have shown the importance of molecular tilting and order-disorder mechanisms to resolve cooperative behaviors[20–22]. The switching of electronic properties coupled with the rapid mechanical effects of cooperative phase transitions presents a wide design space for novel organic electronics such as thermally activated actuators and highly flexible devices[9,23].

However, studies involving structural transitions in n-type counterparts (necessary for full logic design) have lagged behind. N-type semiconductors require relatively low lying LUMO levels for efficient electron transport, which are difficult to stabilize under ambient conditions[24–27]. One route to design stable n-type molecules involves forming a quinoidal structure using strong electron withdrawing groups such as cyano groups and attaching long alkyl chains to the core. The cyano groups break the aromaticity of the conjugated core and planarize the molecule by stabilizing the quinoidal resonance structure that exhibits double bonds between the rings. On the other hand, alkyl chains are widely used to impart solution processability and helps prevent reactions with water and oxygen in the solid state[24,28]. At the same time, these quinoidal cores stabilize an exotic biradical ground state that exists in equilibrium with the quinoidal form[24,28–30]. The presence of biradicals results in interesting spin-spin interactions[31,32] between the conjugated cores and, in extreme cases, results in formation of "pancake" bonding[33]. These unique interactions are also tunable through slight changes in environment (concentration, temperature etc.). Along with intermolecular interactions, biradical formation appears to have self-doping effects capable of tuning the charge carrier densities and the resulting electronic properties[12,28,34,35]. While both the quinoidal core and alkyl chains are key to engineering n-type semiconductor molecules, the effect on structural transitions is still unexplored.

In this work, we report an intriguing coexistence of both cooperative and nucleation and growth type mechanisms during polymorphic transitions in single crystals of 2DQTT-o-B, one of the highest performing n-type organic semiconductors to date[24,36]. The I-II phase transition, which exhibits cooperativity, is accompanied by a significant conformational change of the alkyl side chains, showing the importance of side chain flexibility in accessing cooperative phenomena. In stark contrast, the II-III transition shows a nucleation and growth mechanism driven by an increase in core interactions based on biradical formation in combination with disordering of the alkyl side chains. Further, the phenomenon of biradical interactions directly inducing a polymorphic phase transition has not been reported before. Finally, we demonstrate the use of cooperative behavior through a thermally actuated switching device, a functionality unavailable to a nucleation and growth type mechanisms.

## Results

### Polymorph transition behaviors

Crystals of 2DQTT-o-B (Fig. 1a) were either grown via dropcasting from a 1:1 para-dichlorobenzene and decane mixture or slow evaporation from a 1:1 dichloromethane and ethyl acetate mixture to form both small (50–300 µm) and large (1–3 mm) crystals, respectively (see the "Methods" section). Single crystal X-ray diffraction showed crystals of 2DQTT-o-B packed in a $C_{2/c}$ unit cell with 1-D π-stacks which was designated polymorph I[36]. Based on the Bravis, Friedel, Donnay, and Haker (BFDH) morphology (Supplementary Fig. 1), we observe the π–π stacking occurs along the long axis of the crystal (Fig. 1b). These π-stacks then pack into layers separated by the alkyl side chains (Fig. 1c), which results in the thin needle-like crystals observed in polarized optical microscopy (POM). We set to investigate two thermally induced structural transitions in single crystals of 2DQTT-o-B using polarized optical microscopy, where we heated and cooled crystals at a constant 5 °C min⁻¹ using a Linkam stage. In order to provide an accurate temperature experienced by the crystals, a thermocouple was used to measure the temperature during this heating process. This provided a calibration between the surface temperature and the temperature reported by the Linkam stage. We observed two reversible phase transitions designated the I-II transition at 164 °C and II-III at 223 °C (calculated based on the calibration). The phase transitions were accompanied by modulation of the brightness and color, reflecting changes in the refractive index of the material (Fig. 1d, e, Supplementary Movies 1–2, 3–4). We observed distinct transition behavior suggesting a cooperative transition mechanism for the I-II transition as opposed to a nucleation and growth mechanism during the II-III transition. Below we discuss evidence from in situ POM in detail and this inference is further supported by in situ GIXD, Raman spectroscopy, and EPR spectroscopy discussed later.

The I-II transition proceeded remarkably fast upon heating (Fig. 2a, Supplementary Movies 1, 2). Across many crystals, the heating transition typically occurred within one frame, less than 0.1 s, providing a lower bound for the propagation speed of at least 2000 µm s⁻¹, implying a cooperative-type mechanism. While difficult to capture under normal circumstances due to the speed, this rapid transition is accompanied by a well-defined phase front propagating through the crystal, distinct from typical diffuse phase boundaries under nucleation and growth mechanisms. During cooling, the sharp phase boundary becomes much more evident due to avalanche behavior, where the phase boundary became pinned during transition (Fig. 2b,

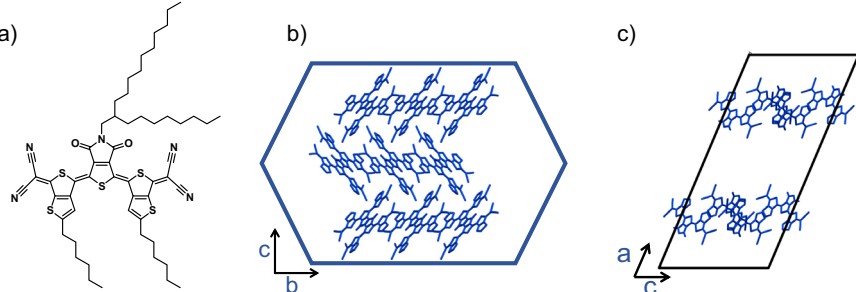

**Fig. 1 | Molecular structure and crystal structure. a** 2DQTT-o-B molecular structure. Crystal packing of polymorph I in the crystal viewed from **b** the top of the crystal (down *a*\*) and **c** along the *b*-direction. The crystal shape is outlined in blue and the unit cell is outlined in black.

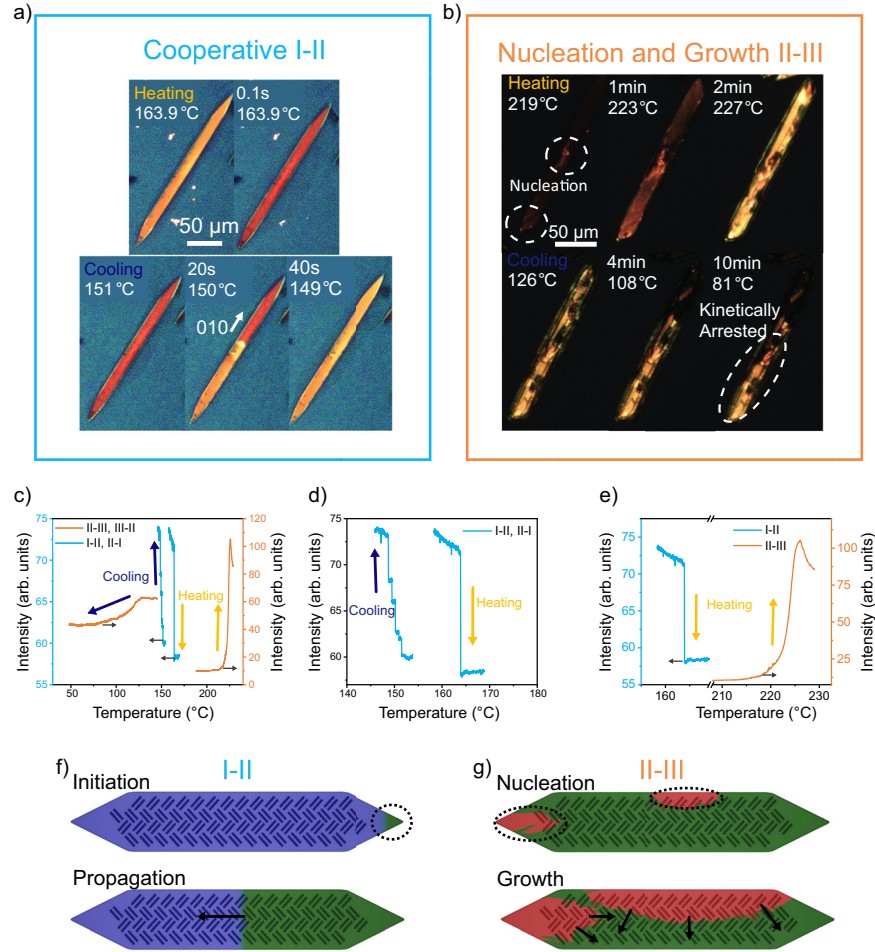

**Fig. 2 | Phase transition kinetics.** In situ POM showing the I-II (**a**) and II-III (**b**). Intensity changes of the crystals observed under in situ POM (**c**), we observe the avalanche behavior of the I-II transition (**d**) and see the stark difference in kinetics (**e**) between the I-II and II-III heating transitions. The dark blue and red lines indicate the I-II and II-III transitions, respectively, and the yellow and blue arrows indicate the heating and cooling transitions respectively. Black arrows indicate the axis for each line. Schematic showing the typical modes for initiation and propagation of the I-II (**f**) and II-III (**g**) transition.

Supplementary Movie 2). Initiation of the phase front typically occurred at the crystal tips, or at cracks within the crystal. Even when the transition initiated at both tips simultaneously, the resulting crystal was a single domain after transition, suggesting a close orientational relationship between the two phases. This is consistent with our earlier work by Chung and Diao et al.[37] which showed defects facilitated phase boundary initiation in similar transitions of ditBu-BTBT followed by cooperative propagation.

Using python-based image analysis, we were able to select individual crystals and track the progression of the phase transition by calculating an average intensity for each frame. From these videos, we tracked the transitions utilizing the change in pixel intensity (average of RGB values) of the crystal during the in situ POM experiment. By using a mask to identify the crystal location, we calculated an average intensity for the crystal in each image and plotted this change in intensity over temperature (Fig. 2c–e). As mentioned, the avalanches were found during cooling, which is clearly observed as plateaus in the crystal intensity during the temperature changes (Fig. 2c, d). These avalanches follow behavior seen in inorganic martensitic transitions, where defects in the lattice act as energy barriers creating local free energy maximums[11,14,38]. The resulting rough free energy landscape causes the boundary to move in a series of jerks and requires thermal fluctuations to jump over the defect barriers. Observing this avalanche behavior upon cooling suggests the initial heating transition may be reforming defect sites within the crystal causing avalanches in subsequent cycles.

Moreover, the direction of the phase boundary is only observable during these pinning events. In these cases, we observe two main crystallographic directions along which the boundary forms: the (010) and (025) planes both of which are connected to the π-stacking direction. The (010) direction lies perpendicular to the 1-D π-stacks, suggesting the cooperative behavior is transmitted down the π-stacks, similar to a domino effect (Fig. 2f). On the other hand, the (025) plane parallels the conjugated core of the molecule and was indexed to the π-π stacking peak in grazing incidence X-ray diffraction[36]. The cross-hatch structure of the π-stacking results in two mirrored directions within the crystal, following the two-phase boundary directions (Supplementary Fig. 2a–c). This would suggest a cooperative behavior is transmitted along the 1-D π-stack and is inhibited by dislocations present in those stacks.

In contrast, the II-III phase transition (Fig. 2b, Supplementary Movies 3, 4) showed nucleation at several points within the crystal and transformed over several minutes, significantly slower than the I-II transition. During this process, polymorph III increased in brightness under polarized microscopy, reflecting changes in the refractive index. The II-III transition exhibited no clear phase boundary, instead we observed a diffuse spreading of the intensely bright polymorph III. The most common nucleation points are near the edges of the crystal, though cracks and other defects will act as nucleation spots (Fig. 1b) as well. We observe a quite smooth change, with no avalanche-like behavior (Fig. 2c, e). Upon cooling, the reverse III-I transition began at a temperature below that of the I-II transition, exhibiting a hysteresis of

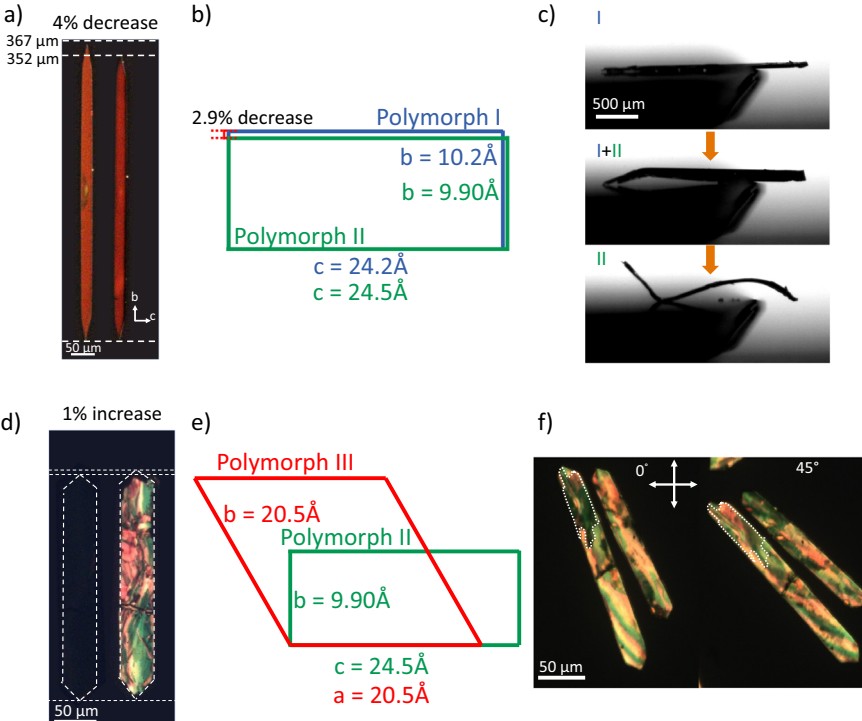

**Fig. 3 | Shape change and thermosalient behavior. a** Crystal shape change associated with the I-II transition and **b** associated in plane unit cell change accounting for the change in crystal length along the b axis of the unit cell (unit cell drawn in the same orientation as the crystal in (**a**)). **c** The resulting thermosalient effect in larger crystals caused by the shape change. **d** Crystal expansion observed in the II-III transition, compared to (**e**) the unit cell changes, where the unit cell is shown in the same orientation as the crystal in (**d**)), and observation of (**f**) birefringence showing multiple domain formation under polarized optical microscopy.

100 °C, an order of magnitude greater than the I-II transition which showed a hysteresis of only 17 ± 5 °C. As a result, this transition may either be a direct III-I transition, or a III-II-I transition (forming polymorph II and then rapidly converting to polymorph I). This matches with observations from DSC using powder samples, showing a hysteresis for the I-II and II-III transitions of 6 °C and 75 °C, respectively[36]. In single crystals, the hysteresis for the II-III transition varied substantially and, in some cases, the III-I transition did not occur during constant cooling, resulting in the kinetic trapping of polymorph III at room temperature. Moreover, the large hysteresis implies either the II-III transition exhibits a much larger energy barrier or smaller driving force than the I-II transition. This suggested a nucleation and growth behavior, shown in Fig. 2g, where nucleation occurred at edges or defects and diffusively spread through the crystal.

## Thermosalient behavior

Upon undergoing the I-II transition, we found the crystal decreased in length by 3.6 ± 1.3% on average (Fig. 3a, Supplementary table 1). This change in length was matched quite closely with the 2.9% decrease in the b-direction of the unit cells obtained through fitting the thin film GIXD diffraction patterns (Fig. 3b), as we previously reported[36]. The discrepancy and large variability in crystal length change is due to crystal bending and substrate pinning that are difficult to discern under the microscope. The pinning of crystals even led to cracking in small crystals of ~100 μm in size (Supplementary Fig. 3, Supplementary Movie 5). This cracking occurs due to the buildup of strain at the phase front caused by the change in shape coupled with crystal-substrate interactions that act to prevent the shape change. In an extreme case, we observed a crystal embedded in a surrounding film that formed during the drying process, which cracked upon transition. This also resulted in the formation of "bright bands" along the phase boundary, visible in Supplementary Fig. 4, which would be consistent with strain induced birefringence we expect from a crystal attempting the I-II

transition while trapped in a solid film. On the other hand, for unpinned large crystals (1–3 mm), we observed a pronounced thermosalient behavior during the I-II transition (Fig. 3c, Supplementary Movie 6). While the crystal will bend during the I-II transition, reversing the transition will not recover original shape, implying a plastic deformation. This may be a result of strain buildup at the phase boundary causing the π-stacks within the crystal to irreversibly slip along the greasy alkyl chain layers.

In the II-III transition, however, we observed a slight expansion to the crystal without thermosalient behavior (Fig. 3d). The change did not conform to the unit cell changes, as polymorph III exhibits a vastly different symmetry and unit cell relative to polymorph II (Fig. 3e) and results from thermal expansion during heating. This reconstructive transition also leads to the formation of multiple grain boundaries within the crystal, breaking single crystallinity (Fig. 3f). Formation of these domains mitigate strain build up preventing any thermosalient motion from occurring due to the crystal expansion. These observations are consistent with the understanding that the nucleation and growth mechanism does not incur shape change owing to a single-to-polycrystalline transition[39]. This stands in contrast to the cooperative martensitic transition that preserves single crystallinity and thus exhibits shape reversibility corresponding to the unit cell change.

## Molecular origin of transition mechanisms

Uncovering the origins of the transition behaviors requires understanding the molecular and structural modifications during the phase transitions. To that end, we performed in situ grazing incidence X-ray diffraction (GIXD) during thermally triggered polymorph transitions. A GIXD video of the I-II transition was obtained by continuous heating at 5 °C min⁻¹ and capturing the diffraction pattern every 5 s (Supplementary Movie 7). The unit cells for polymorphs I and II were extracted from the GIXD patterns via a regression fitting the diffraction pattern as discussed in previous work[36]. This fitting for polymorph I also

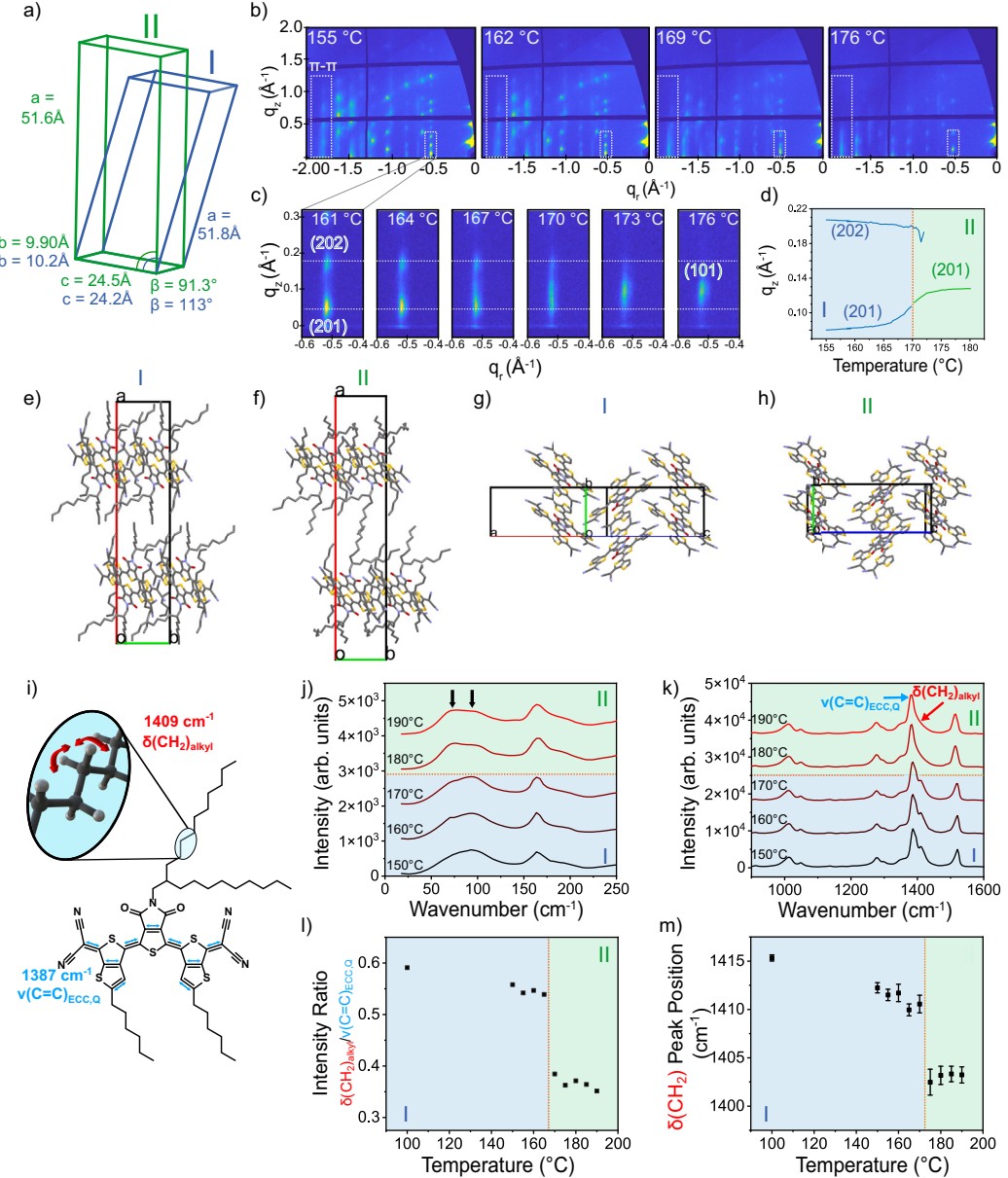

**Fig. 4 | Molecular origins of I-II transition.** Unit cell changes from polymorph I (blue) to polymorph II (green) (**a**). Frames from Supplementary Movie 7, during in situ GIXD (**b**). Changes to the (10) Bragg rod (**c**) reflecting the β-angle shift and (**d**) plot of $q_z$ peak position during the I-II transition. Packing of polymorphs I (**e**) and II (**f**) along the c-direction showing increased core tilt and reduced alkyl chain interdigitation. Polymorphs I (**g**) and II (**h**) along the a* direction with similar 1D π-stack motifs. Schematic of the CH$_2$ deformation mode (δ(CH$_2$), red) and quinoidal ECC (ν(C=C)$_{ECC,Q}$, light blue) (**i**). Raman spectra intensity shifts in phonon region (**j**) and core C=C stretching region (**k**) with increasing temperature. Decrease in the intensity ratio of δ(CH$_2$) to ν(C=C)$_{ECC,Q}$ (**l**), consistent with alkyl chain reorientation during transition. δ(CH$_2$) peak position redshift at the transition temperature (**m**). Error bars reflect peak fitting error. Orange lines indicate transition temperatures.

matched the single crystal unit cell obtained at room temperature[36]. During the I-II transition, we observed minor changes to the unit cell, showing peak movements in the $q_z$ direction while keeping the Bragg rods in place and maintaining the monoclinic symmetry (Fig. 4a, b). The largest extent of peak shift in the $q_z$ position occurred right before the I-II polymorph transition, similar to pretransition behavior observed by Panda and Naumov et al.[8,40] (Fig. 4c, d). This pretransition behavior is typical of pretransition regions of critical phenomena[41], such as in the lattice fluctuations necessary to drive the avalanche behavior[42]. Comparing the fitted unit cells for polymorphs I and II (Fig. 4a), there is a significant increase in the β-angle during the I-II transition, consistent with the observed out of plane shift.

To elucidate the atomistic details of the I-II transition, this transition was simulated adiabatically using semi-empirical

quantum chemistry starting from the unit cell extracted from GIXD for polymorph I. Periodic semi-empirical quantum chemistry calculations at the GFN2-xTB level of theory were performed on the full unit cell, which incorporate the electronic effects that cannot be addressed in typical classical molecular dynamics simulations. The adiabatic transition was simulated starting from the polymorph I lattice parameters and atomic positions, then linearly interpolating to the polymorph II lattice parameters from the SCXRD while continuously relaxing the fractional atomic positions at each step of the interpolation. The validity of the predicted structure of polymorph II was confirmed by comparing the calculated powder pattern, which showed good correspondence to the experimental results (Supplementary Fig. 5a). More importantly, the π-stacking direction in the predicted crystal structure

(Supplementary Fig. 5b) matches the (025) plane predicted from the GIXD pattern.

Comparing the structures of polymorph I and simulated polymorph II (Fig. 4e, f), we observe a change in tilt of the conjugated core of the molecule while the packing maintained similar 1D π-stacks in polymorph II (Fig. 4g, h). A shift of ~12° occurs in the angle of the core with respect to the (100) plane that parallels the silicon substrate of the crystal, from 66° in polymorph I to 54° in polymorph II (Supplementary Fig. 6a, b), indicating the molecule is leaning more towards the substrate which is parallel to the bc-plane in polymorph II. The structures shown in Fig. 4e, f illustrate that this change in tilt angle opens up more volume for the alkyl chains to occupy in polymorph II. While we cannot determine the exact location of the alkyl chains within the structure due to disorder and thermal effects not captured by the adiabatic quantum chemistry simulation, we can see the extra space between layers may lead to reduced interdigitation and more conformational freedom.

To elucidate the driving force behind this change in tilt, we turned to in situ Raman spectroscopy to provide insight into the molecular changes occurring in the I-II transition (Fig. 4d, e). Raman has shown to be a powerful tool for tracking polymorphic transitions and understanding the molecular changes that occur as a result of these transitions[15,43,44]. Under Raman spectroscopy crystals exhibit both low frequency peaks related to the intermolecular phonon vibrations, sensitive to the crystal structure (<500 cm$^{-1}$), as well as intramolecular vibrations related to the chemical environment of the molecule (900–1800 cm$^{-1}$). In the phonon region, we observe two key peaks which are associated with lattice vibrational modes, indicated with black arrows in Fig. 4j. These were identified based on a deconvolution of the peaks at low temperature, shown in Supplementary Fig. 7. We observe a ratio change between these two peaks, as the lowest peak increases in intensity during the I-II transition (Fig. 4j). This is a small effect, indicating modest changes to the lattice packing consistent with our observation of changing molecular tilt from GIXD. Phonon intensity modulation, as opposed to peak shifts or splitting has been shown to be typical in well-studied martensitic type cooperative transitions[45,46]. On the other hand, in the intramolecular vibration region (900–1800 cm$^{-1}$), we observed significant changes to the intensity of the peak at 1409 cm$^{-1}$ relative to the most intense peak at 1387 cm$^{-1}$ indicated by the red and blue arrows in Fig. 4g, respectively and illustrated in Fig. 4e. Based on a combination of DFT calculations (Supplementary Fig. 8b) and literature, we assigned the peak at 1387 cm$^{-1}$ to C=C stretching along the backbone, corresponding to the effective conjugation coordinate (ECC) mode for the quinoidal form[47–49] which we denote as $\nu(C=C)_{ECC,Q}$ (Supplementary Fig. 8c, Supplementary Movies 8, 9); the peak at 1522 cm$^{-1}$ was assigned to other C=C stretching modes in the backbone (Supplementary Fig. 8e, Supplementary Movie 10); the peak at 1772 cm$^{-1}$ was assigned to C=O stretching (Supplementary Fig. 8f, Supplementary Movie 11). While the peak at 1409 cm$^{-1}$ may correspond to core stretching (Supplementary Fig. 8d, Supplementary Movie 12), this was assigned to CH$_2$ deformation in the alkyl chains denoted as $\delta(CH_2)$ based on well-established literature[50–53] on alkyl chain vibrations. Along with these peaks, we also tracked the peaks at 1025 and 1063 cm$^{-1}$ which were assigned to C–C stretching of the trans $\nu(C-C)_T$ and gauche $\nu(C-C)_G$ isomers in the alkyl chains. The alkyl chain peak assignments ($\delta(CH_2)$, $\nu(C-C)_T$, $\nu(C-C)_G$) were confirmed based on Raman spectroscopy measured on a similar molecule with shorter side chains (Supplementary Fig. 9). A complete explanation for these assignments can be found in the supplementary information.

To investigate the changes in these vibrational modes during the phase transitions, peaks were fit using OriginPro in both the 1000–1100 cm$^{-1}$ region and 1350–1600 cm$^{-1}$ region (Fig. 4g). Representative fitting results for each polymorph can be found in Supplementary Fig. 10. Peak line shape was selected as either Gaussian or Lorentzian based on whether the vibrational mode was associated with the quinoidal core or alkyl chains, respectively. Normally, the line shape in solid crystals is expected to follow a Gaussian curve; however, peaks associated with the alkyl chains were fit as Lorentzian due to the disorder, causing more liquid-like behavior[54–56]. In the 1000–1600 cm$^{-1}$ range, which captures the core and alkyl chain vibrations, we note a significant intensity reduction and redshift of the $\delta(CH_2)$ peak at 1409 cm$^{-1}$ during the I-II transition. We plotted the intensity ratio with respect to $\nu(C=C)_{ECC,Q}$ and the peak position to show this sudden drop in intensity and redshift of 7 cm$^{-1}$ in the $\delta(CH_2)$ at the I-II transition temperature (Fig. 3h, i). This trend was repeated across 3 crystals to ensure this change is representative of the I-II transition (Supplementary Fig. 11a, b), though the exact transition temperature varied by 5 °C for each crystal due to avalanche behavior. The redshift is indicative of reduced steric hindrance or an increase in the attractive forces between alkyl chains. The intensity decrease, on the other hand, implies alterations to either the orientation or distribution of vibrational states in the alkyl chains. Taken together, these changes show a substantial reorientation of the alkyl chains occurs during the I-II transition, consistent with reduced interdigitation leading to less steric hindrance between alkyl chains.

For the II-III polymorph transition, we see large changes in the unit cell, as polymorph III appears to form a likely hexagonal crystal system (Fig. 5a), obtained from the GIXD pattern as discussed in previous work[36]. This is consistent with other previous observations which have shown crystals have a propensity to increase in symmetry at higher temperatures[57]. GIXD video of the II-III transition (Fig. 5b, Supplementary Movie 13) showed the significant reduction in observable peaks, as a result of increased symmetry of the hexagonal lattice as well as reduced long-range order as higher q peaks are no longer measurable (to be discussed later). Despite the reduced diffraction of other Bragg rods at higher q values, the π-stacking peak remains intense and well defined in polymorph III, in line with a continued dominant π-stacking motif. The reconstructive transition is further highlighted by observing the complete disappearance of the polymorph II diffraction peaks at the expense of polymorph III peaks (Fig. 4c, d), with no shift in position prior to transition. Unfortunately, the full crystal structure could not be simulated for polymorph III due to the increased thermal disorder of the alkyl chains and potential changes to the molecular structure due to biradical formation (vide infra).

Upon investigation of the II-III transition mechanism via Raman spectroscopy, we observed something quite unexpected: we see the marked weakening of the core stretching peak $\nu(C=C)_{ECC,Q}$ at 1387 cm$^{-1}$, while 2 new peaks appear at higher wavenumbers, 1442 and 1497 cm$^{-1}$ (Fig. 5e, f). This indicated a modification to the conjugated core itself. We initially suspected this may be a degradation of the molecule at such high temperatures. However, the temperature at which the peaks appeared were highly correlated with the II-III transition and, moreover, this peak formation was found to be reversible (Supplementary Fig. 12). Along with this, we also observed a change in the intensity ratio between the peaks located at 1025 and 1063 cm$^{-1}$, previously assigned to the trans ($\nu(C-C)_T$) and gauche ($\nu(C-C)_G$) isomers, respectively. Based on these observations, we hypothesize a transformation from a quinoidal structure to aromatic structure accompanied by decreased order of the alkyl side chains; such quinoidal to aromatic transitions have been observed in similar systems[12,27,29,47].

Previous work on similar quinoidal systems has revealed the presence of a singlet biradical ground state which affects the molecular structure and, by extension, the intermolecular interactions. Moreover, these biradical forms showed a sensitivity to temperature offering the intriguing possibility that the II-III transition occurred alongside a quinoidal to aromatic transition[12,33]. The key change in molecular structure between a quinoidal and aromatic form is the alternation between the single and double bonds along the conjugated backbone, which is described by the bond length alternation (BLA) parameter, the average difference between consecutive single and

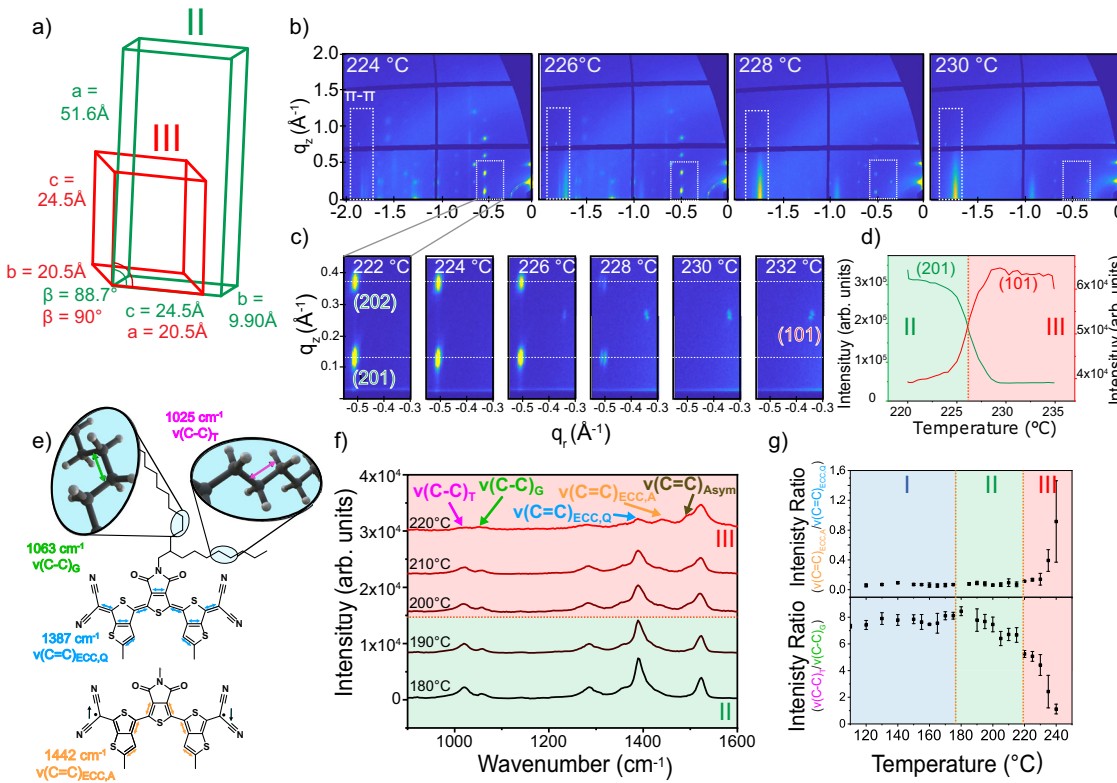

**Fig. 5 | Molecular origins of II-III transition. a** Unit cell changes from polymorph II (green) to polymorph III (red). **b** Frames from (Supplementary Movie 13), showing the diffraction peak evolutions during in situ GIXD. **c** Enlarged view of the (20) (polymorph II) and (10) (polymorph III) Bragg rods and **d** associated plot of $q_z$ peak position during the I-II transition. **e** Main vibrational modes for the peaks assigned to the $\nu(C=C)_{ECC,Q}$ $\nu(C=C)_{ECC,A}$. **f** Evolution of in situ Raman spectra with increasing temperature for the core C−C stretching region. **g** Intensity ratio of aromatic conjugated core to quinoidal core increase during II-III phase transition (Top), and corresponding decrease in the trans to gauche intensity ratio (Bottom). Error bars reflect the standard deviation of 3 measurements in different locations in the crystal.

double C−C bonds. A BLA < 0 defines the quinoidal structure whereas a BLA > 0 defines the aromatic form. Because of this change in bond length, Raman spectroscopy is quite sensitive to these biradical forms, due to the difference in vibrational frequency of the ECC mode in the quinoidal and aromatic systems[12,27,29,47]. We simulated the Raman spectrum for this scenario by forcing the BLA into the aromatic form via substituting the cyano groups for methyl groups on the ends of the molecules and compared this to the spectra obtained from polymorph III (Supplementary Fig. 13a). This allowed us to assign the two new peaks at 1442 and 1497 cm⁻¹ to the ECC of the aromatic form $\nu(C=C)_{ECC,A}$ (Supplementary Fig. 13b, Supplementary Movie 14) and asymmetric stretching modes along the conjugated backbone $(\nu(C=C)_{asym})$ (Supplementary Fig. 13c, Supplementary Movie 15) respectively. The complete explanation for these assignments is found in the supplementary information.

Based on this assignment, we plotted the intensity ratio of the 1442 cm⁻¹ to the 1387 cm⁻¹ peaks, representing the ratio of aromatic to quinoidal forms, as we increased temperature (Fig. 5g, top). We observe a significant increase in the biradical form starting at the II-III transition. Moreover, the intensity ratio of trans to gauche C−C stretching also changed during the transition (Fig. 5g, bottom). As polymorph III formed, this ratio decreases drastically, suggesting significant changes in the alkyl chain dynamics. In previous works, the intensity ratio of these two conformers provide an indication of disorder in the alkyl side chains[50,52,58,59]. As more gauche isomers form in the system, the alkyl chains become bent and twisted within the occupied volume. Disordered alkyl chains (freely rotating C−C bonds) are represented by a ratio of 1, as the chains become free to bend in any direction. In the 2DQTT-o-B system, the intensity ratio of the trans to gauche vibrational modes decreases from 8 to 1 across the II-to-III transition. These results

strongly suggest that the formation of biradical species and alkyl chain disorder are the molecular origins of the II-III phase transition.

To confirm the formation of biradicals, we also simulated and compared the absorption spectra and measured the spin states in the crystals through in situ EPR spectroscopy to directly observe the formation of these biradical states. TD-DFT calculations were used to simulate the UV-Vis spectra of both the quinoidal and aromatic forms of 2DQTT-o-B by constraining the BLA value of each system (see "Methods"). The simulated UV-Vis spectra for the quinoidal structure showed good agreement with the experimental spectra (Supplementary Fig. 15a) though we expect the redshift and vibronic fine features resulting from crystalline packing not to be captured in the simulation. For the aromatic molecule, the simulation showed a significantly red shifted and broadened peak compared to the quinoidal form, similar to the redshift observed in the experimental spectra (Supplementary Fig. 15b). Based on Raman, we calculated the area ratio of aromatic to quinoidal peaks to be 0.92 ± 0.55 in polymorph III, and we see this ratio gradually increases approaching the transition suggesting a mixture of these forms in polymorph III (Fig. 5g). We note the error on these fits were quite large due to temperature effects and it is unclear if this ratio would continue to increase towards biradicals at even higher temperatures. Nevertheless, this resulting mixture may account for a smaller extent of redshift observed than predicted by simulation (Supplementary Fig. 15c, d). Nonetheless, the good qualitative match between simulated and experimental spectra suggests that the biradical aromatic form provides a good model for describing the molecular system in polymorph III.

To understand the nature of this biradical species and the interactions between the dimers in polymorph III, broken-symmetry DFT calculations were performed to assess the biradical characteristics of

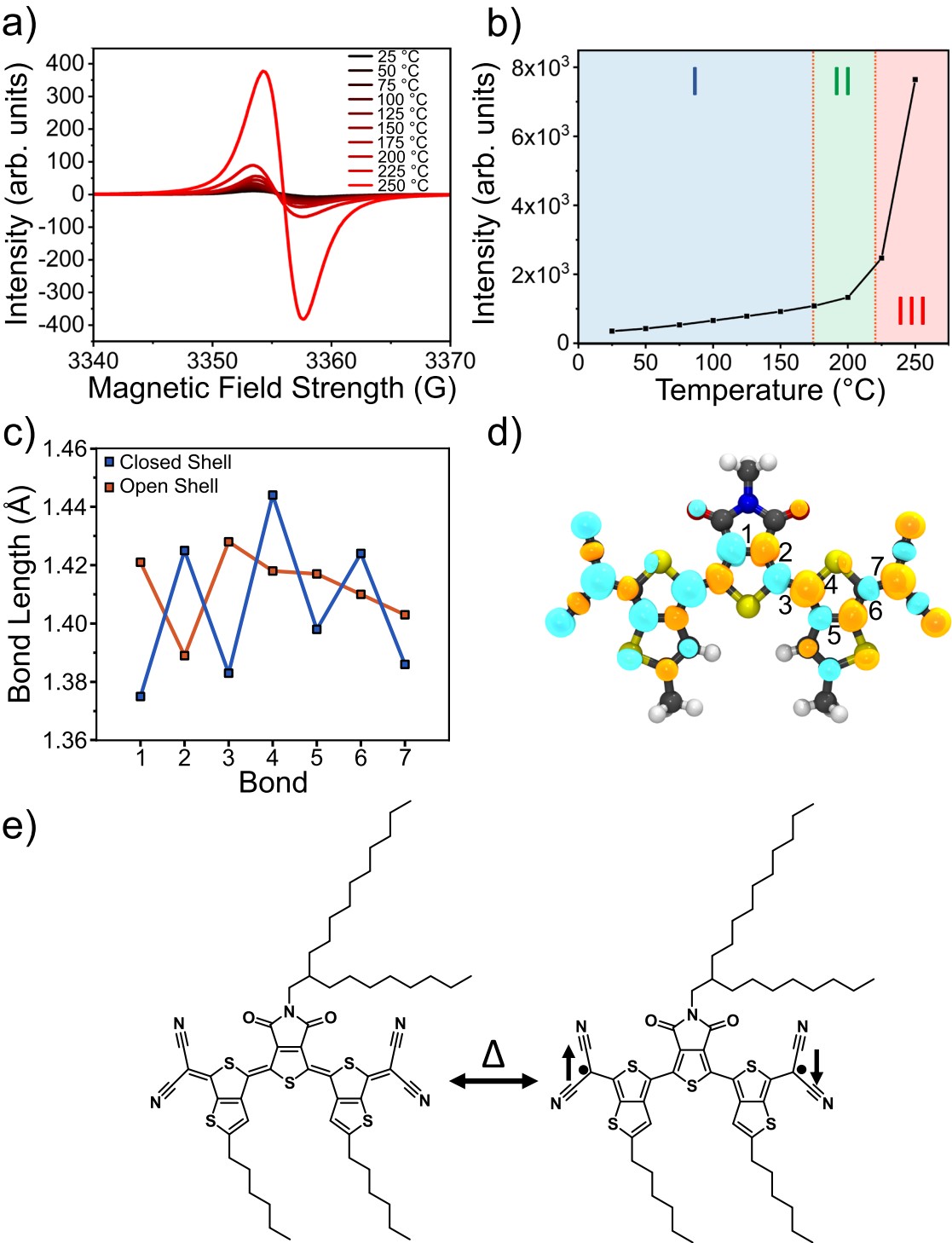

**Fig. 6 | Biradical formation. a** In situ EPR spectra (**b**) doubly integrated intensity across varying temperature showing a substantial increase in number of spins with heating. **c** Bond length comparison between the closed shell and open shell forms and **d** spin densities of the $\alpha$ (cyan) and $\beta$ (orange) spins for the open shell singlet state. The numbering of the bonds in (**c**) is shown in (**d**). **e** Proposed schematic for biradical formation associated with the EPR signal.

the molecule in the open-shell singlet state. The BLA values of the calculated open-shell singlet state and the closed-shell singlet are in good agreement with the proposed quinoidal and aromatic forms, respectively (Fig. 6c). The biradical characteristics were evaluated by (i) the biradical character, (ii) the singlet-triplet energy gap, and (iii) the effective electron exchange interactions. The biradical character y describes the open-shell character of a molecule where the value ranges from 0 (closed-shell state) to 1 (pure biradical state). The results showed that its biradical character (0.35), single-triplet energy gap

(−8.5 kcal/mol), and electron exchange interaction (−4.2 kcal/mol) is indicative of significant open-shell biradical state[12,60], while the positive energy difference between open-shell singlet and closed-shell singlet (2.7 kcal/mol) states confirm that the closed-shell singlet is the pre-ferred ground state, consistent with the biradical character appearing only at elevated temperatures. The spin density of this open-shell singlet state (Fig. 5d) is well distributed over the whole conjugated core, where the carbon linking the cyano groups has the highest density of 0.27.

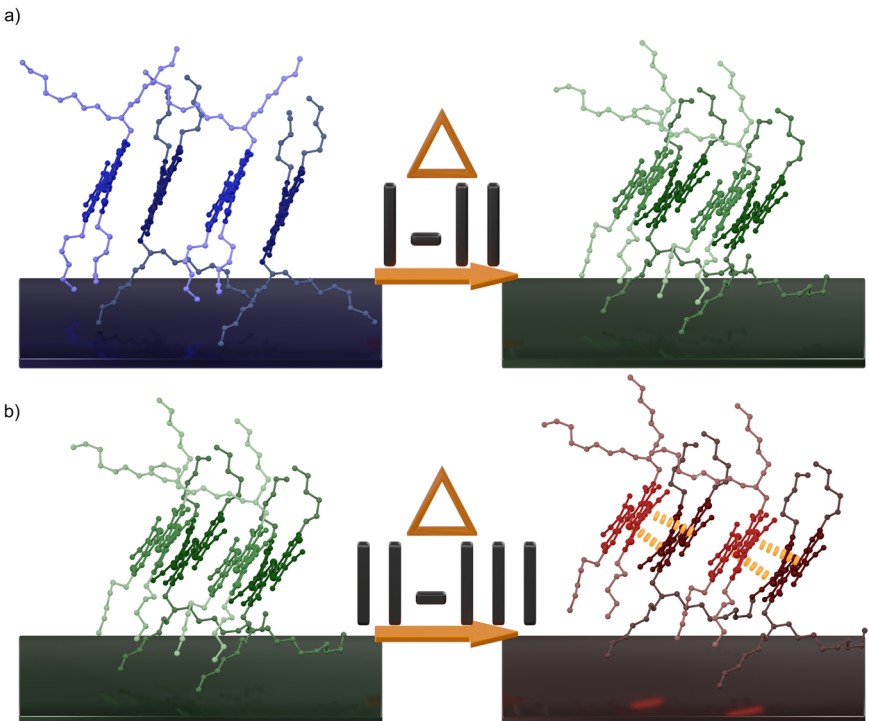

**Fig. 7 | Proposed transition mechanisms.** Schematics of the transition mechanisms for the **a** I-II and **b** II-III transitions. Conjugated cores and alkyl chains are colored separately to improve contrast between molecules.

Finally, as expected from our hypothesis, in situ EPR showed an increasing presence of spins in our system as we increased temperature (Fig. 6a). EPR directly measures the increasing unpaired spins associated with the biradical state (Fig. 6a) and indeed, we see a dramatic increase in the concentration of spins within the material (40x increase) upon reaching the II-III transition (Fig. 6b). At the elevated temperature where the singlet-triplet energy gap is expected to be much smaller, the calculated biradical character at zero Kelvin and the increased EPR intensities observed for the polymorph III suggest that the dimers within polymorph III can form strong spin-spin interactions. In addition, X-ray photoelectron spectroscopy showed the presents of approximately 20% charged species room temperature (Supplementary Fig. 15a, b), consistent with other similar molecules, potentially resulting in charge interactions as well. While distinguishing between these effects at high temperature is difficult, the shift to the open shell biradical form results in new intermolecular interactions. This confirms that the II-III transition not only exhibit a large structural change, but a key part of this transition involves accommodating the abruptly increased concentration of biradical species through a reconstructive structural transition (Fig. 6e).

**Proposed structural transition mechanisms**

Based on the molecular changes described previously, we suggest the I-II cooperative transition occurs via a change in the tilt in the molecule driven by alkyl side chain conformation change (Fig. 7a). Much like stacks of dominos tilting all at once, molecules along the b-axis of the crystal constituting the 1-D π stacks stand up during the I-II transition. In this case, the tipping of the molecule is caused by a change in the interdigitation of the alkyl side chains, which results in the molecule tilting during heating, much like a series of dominos falling over. We suggest engineering of these alkyl side chains may be crucial in facilitating the I-II cooperative behavior. While recently rotation of bulky side chains have been shown to be critical for expressing cooperative behavior in molecular crystals[15,16,18], cooperative polymorph transition has rarely been studied in molecules with alkyl side chain motifs—a

dominant side chain design in organic crystals[23,61,62]. Until now, their role in facilitating cooperative transitions has not been well understood.

The II-III phase transition, however, was driven by the formation of biradical species and is facilitated by increased mobility of the alkyl side chains becoming disordered. At room temperature, the concentration of biradical cores is quite low, meaning the potential for spin-spin interaction is quite low. However, as the temperature increases through the II-III transition, the biradical concentration abruptly increases conducive to forming new spin-spin interactions[27,33]. Typically, in molecules where the biradical ground state is dominant, the major packing motifs are either face-on dimer pairs when unhindered[30,63] or ladder-like stacking[31,64] when steric hinderance prevents the former. In both cases, the high spin centers (where the radical is most localized) will line up and, in some circumstances, form a pseudo σ-bond[12,30,33,63].

One possibility of molecular packing change during II-III transition is shown in Fig. 7b, where molecules form dimers with the nearest neighbor in the original polymorph II structure. Since the crystal packs in these 1-D π-stacks with dimer-like pairs already, this would be a natural progression. While the dimer formation may be plausible, we cannot say with certainty how those dimers then pack to form polymorph III. Regardless of the new core packing motif, to facilitate the II-III also necessitates disordered alkyl chain, indicated by increased gauche conformers. Moreover, this side chain disorder is consistent with the kinetic trapping observed (Fig. 2b and Supplementary Movie 4). As we observe upon cooling to 37 °C, the alkyl chains return to a frozen state, however, there appears to be residual biradicals not yet quenched by cooling, owing to a residual lifetime of the biradical state (Supplementary Fig. 12). As a result, polymorph III is trapped and would not return to polymorph I due to freezing the alkyl chains before the conjugated cores rearrange back into polymorph I that may require quenching of biradical species. However, without the full structure of polymorph III, we are left only to postulate possible molecular packings. Nevertheless, we can suggest the nucleation and growth polymorph transition is the result of increased core–core

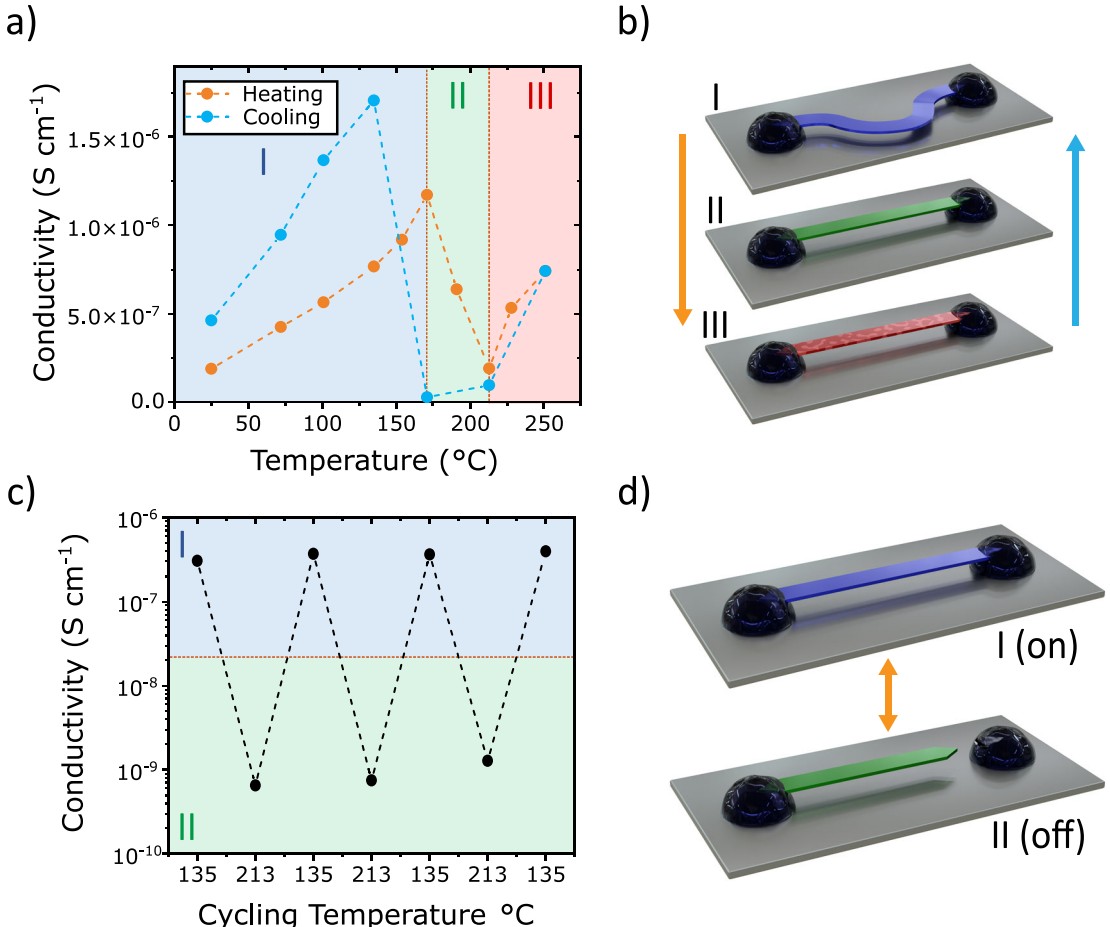

**Fig. 8 | Cooperative transition driven actuator device. a** In situ temperature variable conductivity measurements showing modulation of the conductance based on polymorph and **b** corresponding device architecture showing the crystal attached with PEDOT:PSS to a silicon substrate in polymorphs I, II, and III (blue, green, and red, respectively). **c** Cycling of single crystal actuator device and **d** corresponding device architecture showing the actuator in polymorph I and II states (blue and green, respectively).

interactions triggered by the biradical formation, and the large packing changes are necessarily facilitated by increased molecular mobility from the disordered alkyl side chains.

## Electronic and device properties

We worked to demonstrate the control and design of organic electronics utilizing these transition mechanisms. We fabricated single crystal devices to measure the two-point-probe conductivity of each polymorph (Fig. 8a, b). The crystals were pre-bent fixed at two ends by PEDOT:PSS electrodes. When heated to trigger I-II transition, the crystal would straighten from the pre-bent state due to shape change shown in Fig. 2a. We observed a 6-fold decrease in conductivity at the I-II transition, with good recoverability during cooling albeit with a hysteresis. Moreover, we see a moderate increase in conductivity upon polymorph II-III transition. We suspect this recovery of conductivity may be a result of charge carrier doping through the presence of biradicals, similar to Wudl et al., which showed a self-doping effect[34]. This self-doping process may compete with the reduction in charge transfer pathways due to structural transitions.

We then were able to harness the shape change discovered in the I-II transition for reliable thermal actuation (Fig. 8c, d). For the device, we attached a crystal at both ends using PEDOT:PSS and upon heating, the crystal became dislodged on one side due to the cooperative shape change. This allowed for switching on and off the device by cycling between polymorphs I and II, exhibiting an on/off ratio of 500 and effectively reaching the noise floor for device measurement in the off

state (Supplementary Fig. 16). Because the crystal became detached in the polymorph II state, we observed no conductivity and cooling back to polymorph I reforms the contact between the crystal and the PEDOT:PSS, turning the device back on. This provided quite reliable cycling for several cycles until out of plane bending prevented stable contact between the PEDOT:PSS contacts. While the cyclability could be significantly improved with devices geometries limiting the out of plane crystal bending, the conductivity in the polymorph I state was consistently recovered at least 94% of the original conductivity. This suggests there may little degradation of the electronic properties, likely due to the cooperative nature of the transition, and may be cyclable over many transitions without loss of performance.

## Discussion

We observed both cooperative and nucleation and growth behavior within the 2DQTT-o-B system, allowing for direct comparison of molecular origins of these transitions in a single material. We discovered the I-II transition exhibited cooperative behavior, resulting in a rapid, shape changing transition dominated by avalanches as defects were introduced. The transition was found to be driven by changes in the alkyl side chain packing, as revealed by Raman spectroscopy. This was in striking contrast with the slow, smooth II-III transition showing clear nucleation and growth behaviors. Unlike the I-II transition, the II-III transition was driven by changes in spin-spin interactions through biradical formation which is the observation of biradical interactions triggering a polymorphic phase transition.

We then harnessed cooperative behaviors for the design and application of novel organic electronic switching behavior. Not only were we able to tune the conductivity by orders of magnitude via the structural changes, but we could also use the mechanical effects to sever the connection with one of the PEDOT:PSS contacts. By taking advantage of the mechanical effects of the cooperative shape change, we could switch on and off a single crystal device through rapid temperature actuation. This opens the door for fusing the novel mechanical effects of cooperative transitions with device architecture for new functionality. Ultimately, understanding the origin of these transition mechanisms offers pathways to rationally designing organic semiconductors to access cooperative transitions with these exciting properties.

## Methods

### Materials synthesis

Synthesis of 2DQTT-o-B was done following via previous synthesis methods[24]. All the reactions dealing with air- or moisture-sensitive compounds were carried out in a dry reaction vessel under a positive pressure of nitrogen. Unless stated otherwise, starting materials were obtained from Adamas, Aldrich, and J&K and were used without any further purification. Anhydrous THF, toluene, and 1,4-dioxane were distilled over Na/benzophenone prior to use. Anhydrous DMF was distilled over $CaH_2$ prior to use.

An oven-dried round-bottomed flask was loaded with compound 2-Hexylthieno[3,4-b]thiophene (520 mg, 1.0 mmol) in anhydrous DMF (2 mL) and toluene (2 mL) under nitrogen atmosphere. TPD-2Br$_2$ (0.45 mmol) and Pd(PPh$_3$)$_4$ (21 mg, 0.018 mmol) were added to the flask. The reaction was stirred and refluxed for 2 days under dark. The reaction mixture was then cooled to room temperature and the product was precipitated. Pure compound 1 was collected by filtration and washed with MeOH (20 mL) as a brown solid.

To the resulting compound 1 (0.30 mmol) in CHCl$_3$/DMF (6:1, 3.5 mL) was added NBS (0.12 g, 0.66 mmol) in one potion. The reaction mixture was stirred at room temperature in dark for 2 h. The reaction mixture was washed with saturated NaCl solution (20 mL), saturated NaHSO$_3$ solution (20 mL) and saturated NaCO$_3$ solution (20 mL) successively, and the organic layer was dried over MgSO$_4$. After filtration, the solvent was removed under reduced pressure to afford crude product, which was further purified by silica-gel column chromatography with CH$_2$Cl$_2$/petroleum ether (1:2) as the eluent. Compound 2 was obtained as a red-brown solid.

Sodium hydride (32 mg, 0.80 mmol) was added to a suspension of malononitrile (40 mg, 0.60 mmol) in anhydrous dioxane (10 mL) under nitrogen atmosphere and stirred for 10 min at room temperature. To this mixture was added compound 2 (0.20 mmol) and Pd(PPh$_3$)$_4$ (23 mg, 0.02 mmol), which was then heated to reflux. After 6 h, the reaction was cooled to room temperature, added diluted hydrochloric acid (1 M, 6 mL) and DDQ (91 mg, 0.40 mmol), and was stirred at room temperature for 2 h. The resulting mixture was extracted with CH$_2$Cl$_2$ (3 × 50 mL), washed with brine, and dried over MgSO$_4$. After evaporation of the solvent, the residue was purified by a silica-gel column chromatography with CH$_2$Cl$_2$ followed by recrystallization twice in CHCl$_3$/CH$_3$OH to give 2DQTT-o-b as a dark red solid.

Hydrogen nuclear magnetic resonance ($^1$H NMR, Supplementary Fig. 17) and carbon nuclear magnetic resonance ($^{13}$C NMR, Supplementary Fig. 17) spectra were measured on BRUKER AVANCE 300 and BRUKER DMX 400 spectrometers. Chemical shifts for hydrogens are reported in parts per million (ppm, δ scale) downfield from tetramethylsilane and are referenced to the residual protons in the NMR solvent (CDCl$_3$: δ 7.26). $^{13}$C NMR spectra were recorded at 100 MHz. Chemical shifts for carbons are reported in parts per million (ppm, δ scale) downfield from tetramethylsilane and are referenced to the carbon resonance of the solvent (CDCl$_3$: δ 77.0).

2DQTT-o-B: 135 mg (67%). $^1$H NMR (400 MHz, CDCl$_3$): δ 0.84−0.94 (m, 12H), 1.27−1.48 (m, 44H), 1.82−1.87 (m, 5H), 3.05 (t, $^3$J = 7.6 Hz, 4H), 3.55 (d, $^3$J = 7.2 Hz, 2H), 7.28 (s, 2H, 80% for isomer I), 7.31 (s, 1H, 20% for isomer II), 7.67 (s, 1H, 20% for isomer II); $^{13}$C NMR (100 MHz, CDCl$_3$): δ 14.0, 14.1, 22.5, 22.7, 26.3, 28.8, 29.3, 29.3, 29.5, 29.6, 29.7, 30.0, 30.9, 31.5, 31.9, 31.9, 32.1, 37.0, 43.7, 65.8, 113.3, 113.8, 119.1, 127.5, 128.9, 139.5, 142.6, 150.8, 161.7, 161.8, 163.9.

**Single crystal fabrication.** Single crystals of 2DQTT-o-B were fabricated through a dropcasting method. Solutions with concentrations between 10–15 mg mL$^{-1}$ of 2DQTT-o-B dissolved in a 1:1 mixture of dichlorobenzene and decane were heated to 100 °C. 5–15 μL were dropped onto PTS-treated silicon wafers (SiO$_2$/Si) and allowed to dry overnight. For large crystals (SCXRD and Raman experiments), 2DQTT-o-B was dissolved in a 1:1 mixture of dichloromethane and ethyl acetate at 1 mg mL$^{-1}$ in a 30 mL vial (up to about 15 mL of solution was used). The solution was then capped with parafilm and 1–5 holes were created in the film using a needle. The vial was then placed in a glove bag under nitrogen atmosphere for slow evaporation over a few weeks. The nitrogen in the glove bag was refilled every other day to prevent oxygen from reacting with the molecule and ensure saturation of solvent vapors did not occur.

**Polarized optical microscope.** Single crystals grown under the dropcasting procedure were placed on a Nikon H550S with a high-speed camera (Infinity 1) and heating stage (Linkam 402). The chamber was sealed with a magnetic lid and O-ring during heating. The temperature ramp was kept at a constant 5 °C min$^{-1}$. Videos were recorded at framerates ranging from 1 to 7.5 fps. Video analysis was performed using a python program to obtain an average intensity value of a given crystal at each frame.

**Grazing incidence X-ray diffraction.** GIXD was performed at beamline 8-ID-E of the Advanced Photon Source at Argonne National Laboratory[65]. The data were collected at 10.91 keV on a 2D Pilatus 1 M detector. Films of polymorph I were obtained by solution coating from a solution of 2DQTT-o-B dissolved in chloroform and chlorobenzene at 6 mg mL$^{-1}$ onto SiO$_2$ treated with trichloro(phenyl)silane (PTS). Solution coating was performed at 85 °C at a blade speed of 0.3 mm s$^{-1}$ and a blade gap of 100 μm. These films were annealed under nitrogen atmosphere for 30 min at 100 °C to convert the films to polymorph I. In situ thermal annealing measurements were conducted in a He environment with the sample on a commercial thermal stage (Linkam HFSX350-GI), with the temperature ramped at 10 °C min$^{-1}$ and the exposure taken after equilibration at the target temperature for approximately 5 min. GIXD videos (Supplementary Movies 7, 13) were obtained via taking continuous exposures at a constant heating of 5 °C min$^{-1}$. GIXSGUI software was used to correct for detector non-uniformity, beam polarization and to reshape the 2D data into the representation q$_z$ vs q$_r$[66]. The incident angle was set at 0.14° right above the critical angle for total reflectance of the organic thin film. Unit cells were extracted based on the discussion in the previous paper by Davies and Diao et al., 2021.

**Raman spectroscopy.** A Raman confocal imaging microscope with a 532 nm laser (Laser Quantum Ventus 532 with max power 50 mW) and 50× long working distance objective lens (HORIBA LabRAM HR 3D) equipped with HORIBA Synapse back-illuminated deep-depletion CCD camera was used to collect spectra. Using a 300 g mm$^{-1}$ grating, we used a scan exposure time of between 20 and 60 s. An optical density filter of OD = 0.2 was used (OD = log(power transmission factor)), and no beam damage was observed only at the highest temperatures after prolonged periods of laser exposure (>5 min). To eliminate this effect, each spectrum was recorded in a new position on the crystal to prevent overexposure of any particular area. For variable-temperature

experiments, the samples were collected using a Linkam THMS600 heating stage with a closed chamber. The heating and cooling rate was kept at 10 °C min$^{-1}$. Each temperature was equilibrated until the temperature reading stabilized and the Raman laser was refocused to account for substrate thermal expansion (~5 min).

**Electron paramagnetic resonance spectroscopy.** Crystals of Polymorph I were grown via the slow evaporation method from DCM and EA described in the crystal fabrication above, were dried out and then placed in an EPR tube. The crystal sample was then measured on an EMXplus running the Xenon software with variable temperature (part no. E0000071) for in situ EPR measurements. The temperature was stepped by 25 °C up through the II-III phase transition. The Intensity at each temperature was then doubly integrated and plotted vs temperature.

**Electronic properties.** For single crystal devices, large needle-like crystals were grown from a 1 mg mL$^{-1}$ solution of 1:1 mixture of dichloromethane and ethyl acetate. These crystals were pre-bent and attached to a glass substrate, using conductive PEDOT:PSS. Pre-bending allowed for the devices to be heated through the I-II transition without cracking or breaking the device. The device was measured in air using a Keysight B1500A semiconductor parameter analyzer. The current through the crystal was measured by contacting the PEDOT:PSS with 2 probes and current was obtained as a function of source-drain voltage.

Actuator devices were fabricated by attaching the crystal with PEDOT:PSS without pre-bending the crystal. During the first cycle of the I-II transition, one end becomes detached from the PEDOT:PSS due to the crystal shape change. This device can then be cycled multiple times as once cooled down, the crystal reconnects with the PEDOT:PSS contact.

**DFT calculations.** All calculations were performed with Gaussian16 Rev. B.[67] using molecular models with side-chains replaced by hydrogen for computational efficiency. Ground-state geometries of 2DQTT-o-B with quinoidal and aromatic BLA were optimized at the ωB97X-D/[68] def2-TZVPP[69] level of theory. For calculations on the quinoidal form, the original molecule was used. For calculations on the aromatic form, the cyano-groups were replaced with hydrogens. This allowed for fully relaxed structures to be used in both simulations while capturing the correct bond length alternation observed in each conjugated core.

The UV-Vis absorption spectra were calculated using TD-DFT at the ωB97X-D/def2-TZVPP level of theory based on the 10 lowest singlet and triplet excitations. The fully relaxed quinoidal geometry was used for these calculations. To obtain a relaxed aromatic geometry of 2DQTT-o-B with the cyano-groups present, the bond-length alteration obtained from the hydrogenated structure was constrained while allowing the remaining degrees of freedom to relax.

The broken-symmetry calculations used to characterize the potential for biradical interactions in polymorph III were performed at the B3LYP[70]/6−31G(d,p)[71,72] level of theory to facilitate comparisons with published results. The initial geometry was taken from the experimental structure of polymorph I followed by geometry optimization on an antiparallel configuration of a dimer to approximate the condensed phase. The dimer geometry was optimized with each molecule constrained to a triplet instead of singlet multiplicity to avoid collapse into the closed-shell singlet structure. The large spatial separation of the unpaired spins in the biradical singlet makes it geometrically similar to the triplet. The closed shell and open shell geometries of the monomer were obtained by taking a monomer from the dimer geometry and performing geometry optimization in singlet state with the restricted and unrestricted wavefunctions, respectively.

**Adiabatic relaxations.** The polymorph I-II transition simulations were performed at the semi-empirical GFN2-xTB level[73] using periodic boundary conditions as implemented in DFTB + version 21.2[74]. The adiabatic transition was simulated starting with the lattice parameters of polymorph I using linear interpolation to reach the lattice parameters of polymorph II over the course of five steps. At each step, the fractional atomic positions were allowed to fully relax and used as the initial guess for the subsequent step of the relaxation. This procedure simulates the polymorph transition while maintaining the system at minimum energy along the reaction path.

## Data availability
The authors declare that the data supporting the findings of this study are available within the paper and its supplementary information files.

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

## Acknowledgements

Y.D., D.D., S.K.P., and H.C. acknowledge the Sloan Foundation for a Sloan Research Fellowship in Chemistry and a 3M Nontenured Faculty Award. D.D. acknowledges support of DuPont Graduate Fellowship and A. T. Widiger Chemical Engineering Fellowship. Y.D. and P.K. acknowledge partial support by the NSF MRSEC: Illinois Materials Research Center under grant number DMR-1720633 and NSF CAREER award under Grant No. 18-47828. X.Z. acknowledges the National Key R&D Program of China (2017YFA0204700). B.S. and B.M.S. acknowledge support by the NSF under Grant No. 2045887-CBET. This work was conducted in part in the Frederick Seitz Materials Research Laboratory Central Facilities. Portions of this research were carried out at the Advanced Photon Source, a U.S. Department of Energy (DOE), Office of Science User Facility, operated for the DOE Office of Science by Argonne National Laboratory under Contract No. DE-AC02-06CH11357. D.D. acknowledges Dr. Joseph G. Manion (CGFigures) for his tutorials on using 3D rendering software for scientific illustrations.

## Author contributions

D.W.D. designed and conducted the majority of the experimental portion of the project. Y.D. supervised the project and guided the manuscript writing. S.K.P. fabricated and measured the single crystal devices and assisted with related discussions. D.Y. and X.Z. Synthesized the 2DQTT-o-B material. B.S. and B.M.S. designed and conducted the simulations of the crystal structures, the DFT calculations of the biradical state, and helped with writing the manuscript. S.S. and B.M.S. performed the DFT calculations for the Raman and UV-Vis studies. J.S., H.C., and P.K. assisted with setup and collection of the GIXD videos. H.C. additionally helped with POM video analysis. R.W. performed the collection and helped with the analysis for the high temperature EPR spectroscopy.

## Competing interests

The authors declare no competing interests.
