## [Peer Review File · Nature Communications]

Unraveling Two Distinct Polymorph Transition Mechanisms in One n-Type Single Crystal for Dynamic ElectronicsReviewers' Comments:

Reviewer #1:

Remarks to the Author:

This is a fully revised manuscript with a great number of new studies carried out and new data added in the revision. I am ok with the current version in principle. I also checked the authors' responses to the comments from Reviewer #2, based on my opinion, I think that the authors have also well addressed his/her comments. Before the final acceptance, I have two minor comments:

(1) On page 14, the authors stated "Based on Raman, we expect a mixture of quinoidal and aromatic forms, which may account for a smaller extent of red shift observed than predicted by simulation (Figure S15 c,d)." Can authors offer quantified analysis for each form?

(2) For the device properties, I am wondering the cycling repeatability in two-point-probe conductivity measurements of the material (Figure 7a and 7b).

Reviewer #3:

Remarks to the Author:

This manuscript presents two polymorphic transitions of compound 2DTT-o-B, which show very different behavior attributed to two different mechanisms for transition. I have reviewed an earlier version of this manuscript and since then the manuscript has substantially improved. The most notable improvement is the presentation of more structural information, although obtained indirectly. It helps in the understanding of the transition as well as the interpretation of the Raman results. The claims in the manuscript are more substantiated with evidence. I have a few more comments that are mostly related to the figures.

-page 3 lines 95: This is not the first system that has shown both a cooperative and a nucleation-and-growth transition in one system. In fact, several racemic aliphatic amino acids show this type of behavior.

-Figure 1f-h: the resolution of these panels is very low. Even when zooming in details are hard to see. It is not clear to me what the black arrows in the figures indicate.

Figure 3: again the assignments in panel k are hard to read.

Figure 4: some labels are not so clearly visible, especially the gray and yellow ones.

Figure 5a: The labels in the legend are very small.

Figure 6: What do the gray boxes refer to? I think it would help when the cell axis will be indicated in the graph, when possible.

Page 18: This needs to be cyclable many times for device properties. For how many cycles are "reliable cycling" obtained?

I was surprised to see the large temperature change that is needed to switch between the "on" and the "off" state, which makes it impractical for fast switching, given that the hysteresis for the I-II transition is only 17 degrees for single crystals.

Reviewer #1 (Remarks to the Author):

This is a fully revised manuscript with a great number of new studies carried out and new data added in the revision. I am ok with the current version in principle. I also checked the authors' responses to the comments from Reviewer #2, based on my opinion, I think that the authors have also well addressed his/her comments. Before the final acceptance, I have two minor comments:

(1) On page 14, the authors stated "Based on Raman, we expect a mixture of quinoidal and aromatic forms, which may account for a smaller extent of red shift observed than predicted by simulation (Figure S15 c,d)." Can authors offer quantified analysis for each form?

We appreciate the question, and this was something we definitely wondered ourselves. Based on the deconvolution of the Raman peaks and looking at the area ratio in Fig. 4, it appears the aromatic to quinoidal ratio approaches ~1. However due to temperature effects at the highest temperatures the error bars on this data are quite large and we are not sure if this would continue at even higher temperatures. We have added this value in the text with the caveats of this calculation.

Based on Raman, we calculated the area ratio of aromatic to quinoidal peaks to be 0.92 ± 0.55 in polymorph III, and we see this ratio gradually increases approaching the transition suggesting a mixture of these forms in polymorph III (Figure 4g). We note the error on these fits were quite large due to temperature effects and it is unclear if this ratio would continue to increase towards biradicals at even higher temperatures.

(2) For the device properties, I am wondering the cycling repeatability in two-point-probe conductivity measurements of the material (Figure 7a and 7b).

This is an important point. Indeed, the conductivity after cycling each time to polymorph I was quite consistent. We have included this in the discussion:

We then were able to harness the shape change discovered in the I-II transition for reliable thermal actuation (Figure 8c,d). For the device, we attached a crystal at both ends using PEDOT:PSS and upon heating, the crystal became dislodged on one side due to the cooperative shape change. This allowed for switching on and off the device by cycling between polymorphs I and II, exhibiting an on/off ratio of 500 and effectively reaching the noise floor for device measurement in the off state (Supplementary Fig. 16). Because the crystal became detached in the polymorph II state, we observed no conductivity and cooling back to polymorph I reforms the contact between the crystal and the PEDOT:PSS, turning the device back on. This provided quite reliable cycling for several cycles until out of plane bending prevented stable contact between the PEDOT:PSS contacts. While the cyclability could be significantly improved with devices geometries limiting the out of plane crystal bending, the conductivity in the polymorph I state was consistently recovered at least 94% of the original conductivity. This suggests there may be little degradation of the electronic properties, likely due to the cooperative nature of the transition, and may be cyclable over many transitions without loss of performance.

Reviewer #3 (Remarks to the Author):

This manuscript presents two polymorphic transitions of compound 2DTT-o-B, which show very different behavior attributed to two different mechanisms for transition. I have reviewed an earlier version of this manuscript and since then the manuscript has substantially improved. The most notable improvement is the presentation of more structural information, although obtained indirectly. It helps in the understanding of the transition as well as the interpretation of the Raman results. The claims in the manuscript are more substantiated with evidence. I have a few more comments that are mostly related to the figures.

-page 3 lines 95: This is not the first system that has shown both a cooperative and a nucleation-and-growth transition in one system. In fact, several racemic aliphatic amino acids show this type of behavior.

We appreciate your comment. Indeed, we intended to remove this claim from the paper and missed this part.

-Figure 1f-h: the resolution of these panels is very low. Even when zooming in details are hard to see. It is not clear to me what the black arrows in the figures indicate.

It appears the process of creating the PDF compressed the Figures too much. We have adjusted the process of compression to preserve fidelity. Additionally, we have included labels and adjusted the color choice to reflect that the black/gray arrows are indicating which axes to read, and the colored arrows are indicating direction of temperature change (heating or cooling). Seen in figure 1:

Figure 2. Phase transition kinetics. In situ POM showing the I-II (a) and II-III (b). Intensity changes of the crystals observed under in situ POM (c), we observe the avalanche behavior of the I-II transition (d) and see the stark difference in kinetics (e) between the I-II and II-III heating transitions. The dark blue and red lines indicate the I-II and II-III transitions respectively and the yellow and blue arrows indicate the heating and cooling transitions respectively. Black arrows indicate the axis for each line. Schematic showing the typical modes for initiation and propagation of the I-II (f) and II-III (g) transition.

Figure 3: again the assignments in panel k are hard to read.

Similar to the previous comment, this figure should have appeared at much higher resolution. Additionally, we have adjusted the font to make it more easily readable in the event of compression:

Figure 4. Molecular origins of I-II transition. Unit cell changes from polymorph I (blue) to polymorph II (green) (a). Frames from Movie 7, during in situ GIXD (b). Changes to the (10) Bragg rod (c) reflecting the β -angle shift and (d) plot of q_z peak position during the I-II transition. Packing of polymorphs I (e) and II (f) along the c-direction showing increased core tilt and reduced alkyl chain interdigitation. Polymorphs I (g)

and II (h) along the a^* direction with similar 1D π -stack motifs. Schematic of the CH_2 deformation mode ($\delta(\text{CH}_2)$, red) and quinoidal ECC ($\nu(\text{C}=\text{C})_{\text{ECC,Q}}$, light blue) (i). Raman spectra intensity shifts in phonon region (j) and core C=C stretching region (k) with increasing temperature. Decrease in the intensity ratio of $\delta(\text{CH}_2)$ to $\nu(\text{C}=\text{C})_{\text{ECC,Q}}$ (l), consistent with alkyl chain reorientation during transition. $\delta(\text{CH}_2)$ peak position redshift at the transition temperature (m). Error bars reflect peak fitting error. Orange lines indicate transition temperatures.

Figure 4: some labels are not so clearly visible, especially the gray and yellow ones. We have adjusted the font and color of the labels to make it clearer and should be improved with higher resolution.

Figure 5. Molecular origins of II-III transition. (a) Unit cell changes from polymorph II (green) to polymorph III (red). (b) Frames from (**Movie 13**), showing the diffraction peak evolutions during in situ GIXD. (c) Enlarged view of the (20) (polymorph II) and (10) (polymorph III) Bragg rods and (d) associated plot of q_z peak position during the I-II transition. (e) Main vibrational modes for the peaks assigned to the $\nu(\text{C}=\text{C})_{\text{ECC,Q}}$ $\nu(\text{C}=\text{C})_{\text{ECC,A}}$. (f) Evolution of in situ Raman spectra with increasing temperature for the core C-C stretching region. (g) Intensity ratio of aromatic conjugated core to quinoidal core increase during II-III phase transition (Top), and corresponding decrease in the trans to gauche intensity ratio (Bottom). Error bars reflect the standard deviation of 3 measurements in different locations in the crystal.

Figure 5a: The labels in the legend are very small.

Figure 5a legend has been remade to be larger and more readable:

Figure 6. Biradical formation. (a) In situ EPR spectra (b) doubly integrated intensity across varying temperature showing a substantial increase in number of spins with heating. (c) Bond length comparison between the closed shell and open shell forms and (d) spin densities of the α (cyan) and β (orange) spins for the open shell singlet state. The numbering of the bonds in (c) is shown in (d). (e) Proposed schematic for biradical formation associated with the EPR signal.

Figure 6: What do the gray boxes refer to? I think it would help when the cell axis will be indicated in the graph, when possible.

We are unsure which gray boxes the reviewer means. If they mean the gray planes in Figure 7, they refer to the silicon substrate that held the single crystal device during measurement. We have updated the figure description to make this clearer. However, since we are showing the entire single crystal device, and some crystals are bent or form multiple domains (polymorphs I and III respectively), it is difficult to assign the unit cell directions consistently. Though the pi-stacking direction (b-axis) should align along the length of the crystal as discussed in figure 1.

Figure 8. Cooperative transition driven actuator device. (a) in situ temperature variable conductivity measurements showing modulation of the conductance based on polymorph and (b) corresponding device architecture showing the crystal attached with PEDOT:PSS to a silicon substrate in polymorphs I, II and III (blue, green and red, respectively). (c) Cycling of single crystal actuator device and (d) corresponding device architecture showing the actuator in polymorph I and II states (blue and green, respectively).

Page 18: This needs to be cyclable many times for device properties. For how many cycles are "reliable cycling" obtained?

I was surprised to see the large temperature change that is needed to switch between the "on" and the "off" state, which makes it impractical for fast switching, given that the hysteresis for the I-II transition is only 17 degrees for single crystals.

We were able to have the device cycle for 4 times and recovered the entirety of the conductivity each time. The only reason it failed after that was due to slight bending out of plane which caused the crystal to miss the PEDOT:PSS contact on subsequent polymorph switching. The high recoverability of the conductivity indicates that with a device geometry preventing the out of plane bending (i.e. an enclosed single crystal device) this cycling could be vastly improved. However, this would prevent verification of the actuator mechanism. As for the temperature, indeed that could also be improved through using temperatures closer to the transition point. We selected the temperatures used to clearly identify polymorph I and II states, and certainly in future works that can be optimized more thoroughly.

We have included more discussion about this reliability and what it means for future optimization:

We then were able to harness the shape change discovered in the I-II transition for reliable thermal actuation (Figure 8c,d). For the device, we attached a crystal at both ends using PEDOT:PSS and upon heating, the crystal became dislodged on one side due to the cooperative shape change. This allowed for switching on and off the device by cycling between polymorphs I and II, exhibiting an on/off ratio of 500 and effectively reaching the noise floor for device measurement in the off state (Supplementary Fig. 16). Because the crystal became detached in the polymorph II state, we observed no conductivity and cooling back to polymorph I reforms the contact between the crystal and the PEDOT:PSS, turning the device back on. This provided quite reliable cycling for several cycles until out of plane bending prevented stable contact between the PEDOT:PSS contacts. While the cyclability could be significantly improved with devices geometries limiting the out of plane crystal bending, the conductivity in the polymorph I state was consistently recovered at least 94% of the original conductivity. This suggests there may little degradation of the electronic properties, likely due to the cooperative nature of the transition, and may be cyclable over many transitions without loss of performance.